# Crosslinked PVA/Nanoclay Hydrogel Coating for Improving Water Vapor Barrier of Cellulose-Based Packaging at High Temperature and Humidity

**Feng Gu [1], Wenjing Yang [1], Junlong Song [2], Huining Xiao [3], Wangxia Wang [1,\*] and Zhaosheng Cai [1,\*]**

[1] School of Chemistry and Chemical Engineering, Yancheng Institute of Technology, Yancheng 224051, China
[2] Jiangsu Co-Innovation Center of Efficient Processing and Utilization of Forest Resources, Nanjing Forestry University, Nanjing 210037, China
[3] Department of Chemical Engineering, University of New Brunswick, Fredericton, NB E3B 5A3, Canada
\* Correspondence: wang.w.xia@163.com or wangwangxia@ycit.edu.cn (W.W.); jsyc_czs@163.com or caizhaosheng@ycit.edu.cn (Z.C.)

**Abstract:** Improving the water vapor barrier of cellulose-based packaging in tropical conditions is very challenging for moisture-sensitive products. In this work, we developed a crosslinked polyvinyl alcohol/nanoclay (PVA/NC) hydrogel for paper surface coating. Layered NC and crosslinking can effectively improve the stability of PVA and block the flow paths of water molecules at elevated temperature and humidity. The result indicates that the crosslinked PVA/NC hydrogel coating ($4 \, g/m^2$) can reduce the water vapor transmission rate of copy paper from $1861 \, g/m^2/day$ to $195 \, g/m^2/day$ at $38 \, °C$ (90% RH). The coated paper has an initial contact angle of $108°$ and can maintain hydrophobicity ($>90°$) under direct contact with water for three minutes. A Kit No. as high as 12 and a Cobb No. of 10 were also achieved by the coating.

**Keywords:** cellulose-based packaging; water vapor barrier; crosslinked PVA/NC hydrogel

## 1. Introduction

Microplastic derived from petroleum-based plastic packaging has been found in the human placenta, lungs and blood [1–3]. The transfer of microplastic into the food chain creates a great threat to human health, which drives people to seek alternative materials [4,5]. Paper as a traditional packaging material derived from lignocellulose has been widely used in our daily life in terms of its favorable biodegradability, renewability and recyclability [6]. However, the poor barrier property due to its hydrophilicity and porosity highly limits its application, especially in high temperature and humidity conditions [7,8]. In these conditions, the water vapor barrier is the most challenging aspect of paper packaging for moisture-sensitive products [9].

Water vapor can easily absorb on or transmit through the hydrophilic fiber layer [10]. Many efforts have been made to block water vapor transmission. A surface coating with a good film-forming polymer is the most common protocol, i.e., nanocellulose, chitosan, alginate, starch, polyacrylic acid and polyvinyl alcohol [11–14]. Polyvinyl alcohol (PVA, $[C_2H_4O]_n$), with its better barrier performance, was widely used as a coating agent of paper packaging [15]. Layered nanoclay in the coating formulation can create impermeable physical barriers and tortured channels, which can further enhance the barrier performance of cellulosic paper [16]. The introduction of impermeable clay platelets into polymers generated a labyrinth within the structure that significantly delayed the passage of water molecules through the sample. The permeability of the polymer/nanoclay composites is ~2 orders of magnitude lower than that of pure matrices [17]. According to work reported by Chou et al. [18], PVA films with nanocomposites exhibited an enhanced water vapor barrier property owing to the hydrogen bonds and van der Waals forces between the substrate and the fillers, as well as layered filler structure.

However, hydrogen bonds and van der Waals forces between the substrate and the fillers can be easily broken by water molecules. The moisture stability of the PVA/clay composite needs to be improved, especially at high temperature and humidity, i.e., tropical conditions [19]. In tropical conditions, the water vapor has a higher diffusion coefficient, thus promoting its transmission in a cellulose fiber network or composites. Therefore, lowering water vapor permeability in tropical conditions is still a challenge when it comes to developing green-based packaging materials [20]. Herrera's study indicated that crosslinking could improve the moisture stability of the coating layer [21]. The water vapor permeability was reduced by 60% with a crosslinked nanocellulose coating.

Herein, we developed a crosslinked PVA/NC hydrogel possessing a good film-forming property and moisture stability for the paper surface coating to overcome the water-induced swelling, and to improve the water vapor barrier at high temperature and humidity. The influence of layered nanoclay and the cross-linking of the PVA coating on the water vapor barrier were investigated. The morphology, water resistance and grease barrier of coated paper were also studied.

## 2. Materials and Methods

### 2.1. Materials

Polyvinyl alcohol (PVA, Mw 89,000–98,000, 99+% hydrolyzed), nanoclay (NC, hydrophilic bentonite, particle size $285.6 \pm 12.2$ nm, zeta potential $-41.70 \pm 1.04$ mV), and sodium tetraborate (Borax, $Na_2B_4O_7$) were purchased from Sigma-Aldrich (St. Louis, MO, USA). PaperPlex Supreme copy paper ($75$ g/m$^2$, Kuching, Malaysia) was used as base paper in this study.

### 2.2. Nanoclay Exfoliation

In order to ensure a sufficient delaminated and nanosized structure of the nanoclay (NC) platelets, the clay powder was dispersed in water in a concentration of 3% using an ultrasonic mixture at 20,000 rpm for 2 h. The mixture was further mechanically stirred for 2 days at 3000 rpm, and then left for another 2 days to settle the non-delaminated nanoclay. Stable dispersed nanoclay was collected and stored in the refrigerator. The concentration of nanoclay suspension was determined before further application.

### 2.3. Coating Formulations Preparation

A PVA aqueous solution (10%, 5% and 2%, *w/w*) was prepared by dissolving the PVA powder in water at 90 °C. The transparent PVA solution (10%, *w/w*) was cooled down to room temperature and applied on the paper coating. Sufficiently delaminated nanoclay was then added into the PVA solution (5%, *w/w*) and magnetically stirred for 24 h to prepare the PVA/NC coating mixture. For the crosslinked PVA/NC hydrogel coating, the nanoclay was dispersed into the PVA solution (2%, *w/w*) under constant magnetic stirring for 24 h, and then PVA was crosslinked with borax at 90 °C for 2 h. The amount of NC in the PVA solution was 5% (*w/w*). The PVA solution (2%, *w/w*) crosslinked with borax was applied as a PVA gel coating.

### 2.4. Coating Process

The PVA, PVA gel, PVA/NC, and PVA/NC gel coating were applied directly to the surface of the copy paper using a bar-coater (K303 Multi-coater, RK Print Coat Instruments Ltd., Litlington, UK). A Mayer rod No. 4 was used with a coating speed of 2.0 m/min (Figure 1). Then, the coated paper samples were dried in an ambient temperature to remove 80%~90% of the moisture followed by drying in an oven at 60 °C overnight to ensure that it was completely dry. Oven-dried coating papers were weighed to calculate the coating weight.

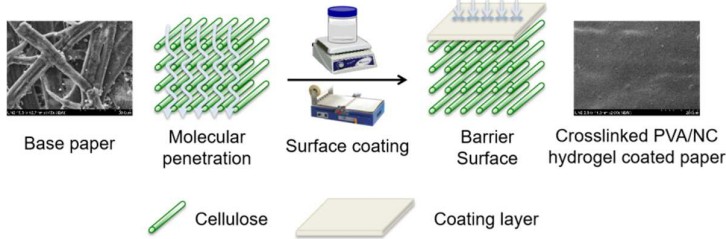

**Figure 1.** Scheme of coating application on paper surface with a Mayer bar.

### 2.5. Nanoclay and Coated Paper Characterization

Surface morphologies of the original and coated paper were observed by SEM (JSM 7600F, JEOL, Tokyo, Japan). Morphologies of NC and PVA/NC film were characterized by an Atomic Force Microscopy (AFM, Nanoscope IIIa, Veeco Instruments Inc., Santa Barbara, CA, USA). The X-ray diffraction (XRD, Bruker D8 Advance spectrometer, Karlsruhe, Germany) was used to determine the interlayer distance of layered nanoclays in a PVA matrix. The diffractometer was equipped with a two-circle (theta-theta) goniometer housed in a radiation safety enclosure. The X-ray source was a sealed in a 2.2 kW Cu X-ray tube and maintained at an operating current of 40 kV and 25 mA. Interlayer distances were calculated by Bragg's Law: $2d \sin \theta = \lambda$, where d is the interlayer distance, $\theta$ is the diffraction angle and $\lambda$ is the wavelength of the X-rays. The X-ray powder diffraction patterns from the samples were run using a Philips X'PertMPD diffractometer (PANalytical, Amsterdam, Netherlands) using Cu X-ray tube ($\lambda$ = 1.542 å).

### 2.6. Barrier Properties of Coated Paper

The water vapor transmission rate (WVTR) was determined according to TAPPI T448om-09, which was carried out at 23 °C (50% RH) and 38 °C (90% RH). The result was calculated by the weight changes per square meter per day [10]. The water contact angle (WCA) was measured by a versatile optical tensiometer (Theta Attension Tensiometer, Attension/Biolin Scientific, Espoo, Finland). The water spreading process throughout the paper was tested and recorded with a dynamic contact angle. Grease resistance was conducted by a Kit test according to the TAPPI standard (T559 pm-12). The water absorptiveness of the coated paper was measured by a Cobb test during 120 s according to the TAPPI standard (T441 om-04). At least three measurements were made for each test and the average values were reported.

## 3. Results and Discussion

### 3.1. Water Vapor Barrier of Coated Paper

Water vapor transmission rates (WVTR) of PVA and PVA gel coated paper at 23 °C (50% RH) and 38 °C (90% RH) are illustrated in Figure 2. The WVTR of original copy paper at 23 °C (50% RH) was 512 $g/m^2/d$. This value can be decreased to 21 $g/m^2/d$ with 4 $g/m^2$ PVA coating, whereas it only needs 1 $g/m^2$ PVA gel coating to achieve the same level. This might be due to the crosslinked PVA gel showing a better stability, which can form a dense coating layer on the paper surface with low loading at room temperature. However, the crosslinked PVA layer is not stable enough at an elevated temperature and humidity. With a higher diffusion coefficient, the water molecules can easily transmit through the cellulose layer and PVA coating layer. WVTR of raw paper is as high as 1861 $g/m^2/d$ at 38 °C (90% RH), which can only be decreased to 510 $g/m^2/d$ with a 4 $g/m^2$ PVA gel coating. Similar results were reported by Sebastien et al. [22]: WVTR of paper board coated with 10 $g/m^2$ PVA was reduced from 390 to 12 $g/m^2/d$ at 23 °C (50% RH). Regarding the change of humidity, the WVTR of PVA coated board can only be decreased from 1016 to 671 $g/m^2/d$ at 23 °C (85% RH). Song et al. [20] also reported that the PLA coating (25 $g/m^2$) only can lower the WVTR of the hand sheet from 2700 $g/m^2/d$ to 600 $g/m^2/d$ at 38 °C (90% RH).

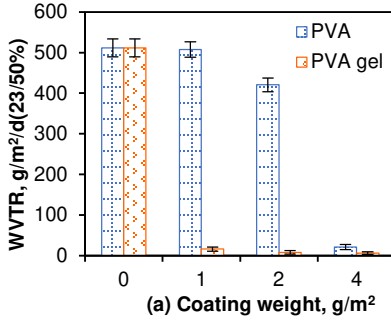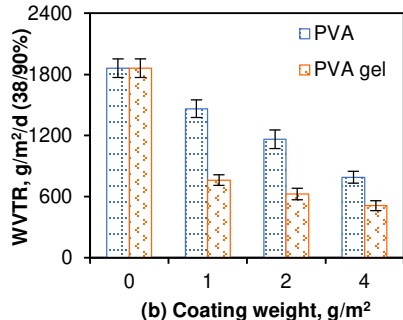

**Figure 2.** The WVTR of PVA and PVA gel coated paper at different temperatures and humidity. (**a**) 23 °C, 50% RH, (**b**) 38 °C, 90% RH.

### 3.2. Effect of Layered Nanoclay Addition and Crosslinking on Water Vapor Barrier

Morphology of delaminated nanoclay is shown in Figure 3a. From the height profiles in Figure 3b, we could find the nanoclay has thickness of 1 nm and width of 100 nm. Well-dispersed nanoclay in a PVA solution can be observed from Figure 4a. The XRD patterns of pure NC and crosslinked PVA/NC film are presented in Figure 4b. According to Bragg's formula (2d sin θ = λ), NC exhibits a typical X-ray diffraction peak at 2θ of 6.7, which signifies that the d-spacing between the interlayers of NC is 1.32 nm. Compared with NC, the broad peak at 6.7 was replaced by a sharp peak located at 5.3 for the crosslinked PVA/NC film. Moreover, the gallery distance (d-spacing) increased to 1.68 nm compared with the pure NC. This is due to the intercalation of PVA chains into the galleries between individual NC platelets in which sheets remain individually exfoliated without any aggregation in the polymer matrix [23].

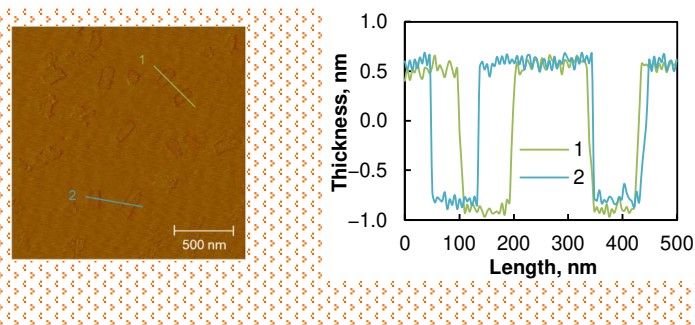

**Figure 3.** The AFM image (**a**) and height profiles (**b**) of delaminated nanoclay.

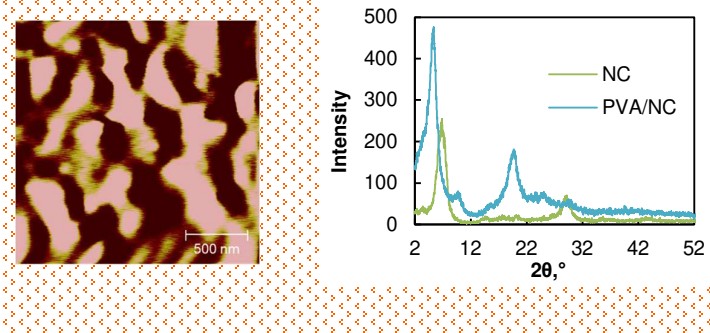

**Figure 4.** The AFM image (**a**) and XRD (**b**) of crosslinked PVA/NC film.

Figure 5 shows the schematic and surface morphology of original and coated paper. As can be seen, the porous structure is obvious for uncoated paper, which is created during paper formation and provides the pathway for molecular transportation (Figure 5a). A PVA gel coating can fill the pores and generate an even surface on base paper to block most channels for molecular transmission at 23 °C (50% RH, Figure 5b). However, without the

support of NC, a PVA gel can hardly form a continuous film on the paper surface to fill the pores between fibers at elevated temperature and humidity (Figure 5c). Meanwhile the well-dispersed NC in the PVA matrix with crosslinking ("crosslinked fiber−brick composite" model) can effectively cover the pores among fibers at 38 °C (90% RH), as shown in Figure 5d.

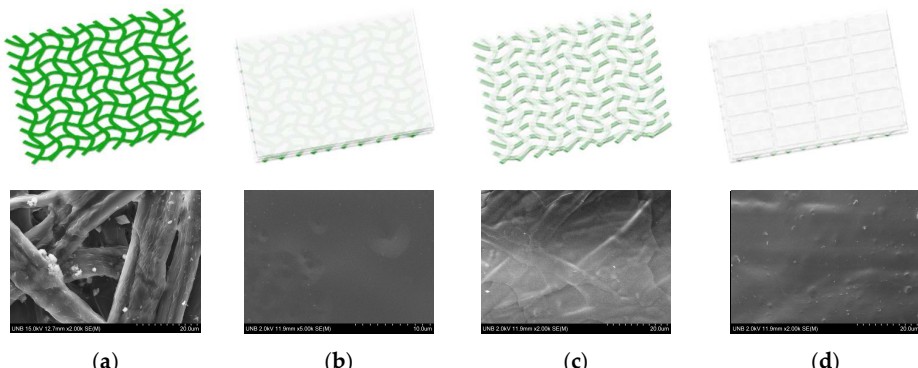

|  (a) | (b) | (c) | (d) |

**Figure 5.** Schematic and surface morphology of original and coated paper. Original paper (Scale: 20 μm, (**a**)). PVA gel coated paper (23 °C, 50% RH, scale: 10 μm, (**b**)). PVA gel coated paper (38 °C, 90% RH, scale: 20 μm, (**c**)). PVA/NC gel coated paper (38 °C, 90% RH, scale: 20 μm, (**d**)).

Nanoclay and crosslinking endow a PVA/NC gel coating layer with better stability, which can be highly effective at blocking the passage of water molecules at elevated temperature and humidity. As a result, a crosslinked PVA/NC gel helps reduce the WVTR by 90% from 1861 g/m²/d to 195 g/m²/d at 38 °C (90% RH), as illustrated in Table 1. In order to eliminate the thickness effect, the corresponding water vapor permeability values were calculated to be 2.06 to 0.24 × 10⁻⁵ g/m/d/Pa [24]. Neither the PVA gel (without NC) nor the PVA/NC (without crosslinking) can achieve a satisfactory water vapor barrier in this tropical condition. This is mainly attributed to the "crosslinked fiber−brick composite" structure, which creates a moisture stable linkage and tortuous diffusion pathway [25]. The reduction of WVTR by a 4 g/m² crosslinked PVA/NC gel coating is comparable with a 40 g/m² of PLA composites coating containing 1% of modified nanofibrillated cellulose [20]. The limited WVTR at high temperature and humidity is very important for bio-based packaging for fruit and vegetables, which are usually stored at a relative humidity of 85%~95%.

**Table 1.** Nanoclay addition on WVTR (g/m²/d, coating weight: 4 g/m²).

| Exposed Condition | Raw Paper | Coating Layer | | | |
| --- | --- | --- | --- | --- | --- |
| | | PVA | PVA Gel | PVA/NC | PVA/NC Gel |
| 23 °C, 50% RH | 512 (26) | 21 (6) | 6 (3) | 5 (2) | 2 (1) |
| 38 °C, 90% RH | 1861 (53) | 789 (59) | 510 (48) | 443 (39) | 195 (29) |

PVA coated paper also exhibited an excellent grease and water barrier. As shown in Table 2, the Kit No. of all types of coated paper is 12 with the Cobb No. being less than 15. The coated surface showed hydrophobicity with an initial water contact angle over 100°. Among the different type coating layers, the crosslinked PVA/NC hydrogel coated paper has the best water resistance with an initial contact angle of 108° and maintained hydrophobicity (>90°) under direct contact with water for 3 min (Figure 6). It indicates the water spreading rate on the paper surface was highly reduced, compared with a rapid water contact angle decrease of base paper from 95° to less than 20°. The improved water resistance makes crosslinked PVA/NC hydrogel coated paper an eco-friendly barrier packaging for moisture sensitive products.

**Table 2.** Grease and water barrier of coated paper (coating weight: 4 g/m$^2$).

| Coating Layer | Kit No. (15 s) | Cobb No. (120 s), g/m$^2$ | IWCA, $^\circ$ |
|---|---|---|---|
| PVA | 12 | $14 \pm 2$ | $101 \pm 9$ |
| PVA gel | 12 | $13 \pm 2$ | $109 \pm 6$ |
| PVA/NC | 12 | $12 \pm 2$ | $105 \pm 8$ |
| PVA/NC gel | 12 | $10 \pm 2$ | $108 \pm 5$ |

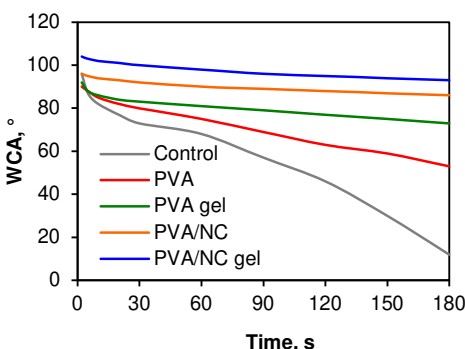

**Figure 6.** Water resistance of coated paper.

## 4. Conclusions

The effects of nanoclay and cross-linking of a polyvinyl alcohol coating on a water vapor barrier of cellulose-based packaging were systematically investigated in this work. The results indicated that a crosslinked PVA/NC hydrogel coating (4 g/m$^2$) can reduce the WVTR of copy paper from 1861 g/m$^2$/day to 195 g/m$^2$/day by 90% at 38 $^\circ$C (90% RH). Improved water and grease barriers were also achieved by the hydrogel coating with Kit No. of 12 and Cobb No. of 10. The findings suggest the barrier-enhanced packaging by crosslinked PVA/NC hydrogel coating has great potential for bio-based applications considering the biodegradability, biocompatibility, and recyclability.

**Author Contributions:** Conceptualization, W.W.; Data curation, F.G. and W.Y.; Formal analysis, W.Y.; Investigation, W.W.; Methodology, F.G. and W.Y.; Writing—original draft, F.G.; Writing—review & editing, J.S., H.X., W.W. and Z.C. All authors have read and agreed to the published version of the manuscript.

**Funding:** The authors are grateful for the financial support from the National Natural Science Foundation of China (No. 21908188, 22008206 & 32071706), the Natural Science Foundation of Jiangsu Province (No. BK20181051 & BK20181052), the NSERC and Asia Pulp & Paper Canada.

**Institutional Review Board Statement:** Not applicable.

**Informed Consent Statement:** Not applicable.

**Data Availability Statement:** The authors confirm that the data supporting the findings of this study are available within the article.

**Conflicts of Interest:** The authors declare no competing financial interest.

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
