# Peer review of "Crosslinked PVA/Nanoclay Hydrogel Coating for Improving Water Vapor Barrier of Cellulose-Based Packaging at High Temperature and Humidity"

_coatings, doi:10.3390/coatings12101562_

Round 1

Reviewer 1 Report

The authors present a new study regarding the preparation of crosslinked PVA/ nanoclay hydrogels for improving the water vapor barrier of cellulose-based packaging and bring to attention new insights regarding the influence of high temperature and humidity conditions for such applications.

There are a few suggestions to improve the quality of the work: 

1. The introduction section should include more relevant and recent studies on using PVA/nanoclay systems for such applications

2. Section 2.3 (Methods) should be reformulated to clearly explain the solution types that were prepared.

From the phrase "Sufficiently delaminated nanoclay was then added into the PVA solution (5%, w/w)" it is not clear which solution present this concentration since before it is stated that 10% PVA solution was prepared at 90 Celsius degrees. Maybe a schematic figure would be more explanatory in this section.  

3. Has the concentration of the nanoclay suspension after preparation under ultrasonication been determined? 

Reviewer 2 Report

On request of Coatings, I have revised the manuscript titled “Crosslinked PVA/nanoclay hydrogel coating for improving water vapor barrier of cellulose-based packaging at high temperature and humidity”, by Feng Gu and colleagues.

The main scope of this study was to develop a paper-based packaging having improved vapor barrier capacity also at high temperature and humidity typical of tropical climate to protect moisture sensitive products. To this end, the authors developed a crosslinked polyvinyl alcohol/nanoclay (PVA/NC) hydrogel and used it to coat paper surface. The reported results show that the crosslinked PVA/NC hydrogel coating (4 g/m2) reduces water vapor transmission rate of copy paper from 1861 g/m2/day to 195 g/m2/day at 38 °C (90% RH). Besides, the coated paper was successful in maintaining hydrophobicity (>90°) under direct contact with water for 3 minutes.

Considering for example, that food spoilage by several environmental factors including oxidation, humidity, foodborne bacteria is a critical problem causing high food waste with a negative impact on global economy, studies are desired that search for strategies to improve food packaging mechanical properties, as the present one.

The work is original, the performed experiments are rational, and discussion is clear. The results are interesting.

Unfortunately, there are some issues and incorrectness in the formatting that must be addressed, to make the present manuscript suitable for publication on Coatings.

All the manuscript, including the reference list should be checked to adapt it to the template provided by Coatings.

Please remove spaces between commas and numbers in the references in the text.

Please, use capital letters for all the words in the sub-headings and in the Title (Examples: lines 65, 71, 80, 89, 101, 113 etc.)

Please, check all the abbreviations and specify them at their first mention where necessary.

Please, the captions of Figures not centred but justified.

Introduction is poor and should be improved and enriched with additional references, to provide readers with a more complete background concerning the packaging materials commonly available and the main methods developed to improve their characteristics. To use a Table or Tables could be a smart way to provide such information.

I suggest authors to consider the following Ref. which could be useful to ameliorate and make more complete their introduction:

https://doi.org/10.3390/app11020621

https://doi.org/10.1016/j.foodres.2020.109664

Finally, the chemical structure of PVA should be inserted in the Introduction section.

On these considerations, I suggest Coatings to consider further the present work for publication after the authors have addressed my minor requests.
